# Influence of Hydrodynamic Conditions on the Type and Area of Occurrence of Gas–Liquid Flow Patterns in the Flow through Open–Cell Foams

**DOI:** 10.3390/ma13153254

**Published:** 2020-07-22

**Authors:** Roman Dyga, Małgorzata Płaczek

**Affiliations:** Department of Process Engineering, Opole University of Technology, 45−271 Opole, Poland; r.dyga@po.edu.pl

**Keywords:** open–cell metal foam, aluminum foam, two–phase flow, flow regime map

## Abstract

This paper reports the results of a study concerned with air−water and air−oil two–phase flow pattern analysis in the channels with open–cell metal foams. The research was conducted in a horizontal channel with an internal diameter of 0.02 m and length of 2.61 m. The analysis applied three foams with pore density equal to 20, 30 and 40 PPI (pore per inch) with porosity, typical for industrial applications, changing in the range of 92%–94%. Plug flow, slug flow, stratified flow and annular flow were observed over the ranges of gas and liquid superficial velocities of 0.031–8.840 m/s and 0.006–0.119 m/s, respectively. Churn flow, which has not yet been observed in the flow through the open–cell foams, was also recorded. The type of flow patterns is primarily affected by the hydrodynamic characteristics of the flow, including fluid properties, but not by the geometric parameters of foams. Flow patterns in the channels packed with metal foams occur in different conditions from the ones recorded for empty channels so gas−liquid flow maps developed for empty channels cannot be used to predict analyzed flows. A new gas−liquid flow pattern map for a channel packed with metal foams with the porosity of 0.92–0.94 was developed. The map is valid for liquids with a density equal to or lower than the density of water and a viscosity several times greater than that of water.

## 1. Introduction

Multiphase flow involving gas and liquid phases forms the most common type of flow in industrial processes that apply porous or structural packing. The role of the porous structure can be taken on by open–cell solid foams, including open–cell metal foams. Due to the high pore density (usually above 90%) and completely open structure (large, empty cells connected to each other), there is sufficient space in which liquids can flow relatively freely. Many metal foams have a specific surface area above 1000 1/m. hen these characteristics are combined with high thermal conductivity, as well as high thermal and chemical resistance, metal foams can be considered particularly applicable in processes involving heat and mass exchange. Multiphase flows occur, among others, in evaporators of refrigeration equipment [1,2,3,4,5], solar installations [6,7], steam condensers [8,9], catalytic reactors [10,11,12,13,14] and in column apparatus [15,16,17] fulfilled by metal foams.

From a practical engineering point of view, one of the major design difficulties in dealing with multiphase flows which take place in apparatuses packed with open–cell metal foam is that the mass, momentum and energy transfer rates and processes can be quite sensitive to the geometric distribution of the mixture components within the flow, so depend on the type of flow pattern. For example, in flow boiling, the annular flow is preferred due to the total surface area of heating pipe being covered by liquid film and getting better heat transfer conditions. The analysis of flow patterns that are formed in specific hydrodynamic flow conditions is of great importance in the development of calculation procedures applied to such variables as the pressure drops, void fraction, heat and mass transfer coefficients. This analysis, together with the pressure drop characteristic, is crucial to determining the hydrodynamic flow conditions ensuring the lowest possible energy inputs necessary to multiphase mixture transport.

Previously reported studies related to multiphase flows in channels and mini−channels packed with metal foams have been mainly concerned with the intensification of mass transfer processes in chemical reactors [18] or described the boiling process of different refrigerants and they focused on enhancing the two–phase heat transfer performance [19,20,21,22,23,24].

Tourvieille et al. [18] studied the overall external mass transfer and pressure drop performance in the reactor with a square mini−channel (width of 2 mm) packed with three different metal foams as catalyst supports for gas−liquid−solid reactions under an applied pulsing regime. This regime was observed for a wide range of variations in liquid and gas velocities, *v_L_* = 0.002–0.042 m/s, *v_G_* = 0.008–0.06 m/s, respectively. The pulsing regime was characterized by quasi−stagnant liquid (plugs) that occupied the entire cross−section of the channel and was carried by gas portions flowing with various frequencies. Tourvieille et al. [18] stated that foams in confined channels allow mass transfer rates to be increased by up to 50% in comparison with empty channels due to intensive turbulence of the phases in pulsing flow.

In turn, Bamorovat Abadi et al. [19] investigated the flow boiling of R245fa refrigerant in a horizontal mini−tube with copper metal foam packing (porosity of 20 and 30 PPI). Annular flow (recognized as more favorable for heat transfer processes) and other types of flow considered by the authors as intermittent flow were identified. In the intermittent flow, the steam flows through the metal foam in the form of bubbles that are squeezed through the foam cells. Under certain flow conditions, vapor bubbles form compact clusters, resulting in the slug flow (typical gas−liquid flow in empty channels). The authors of Reference [19] defined the transition line between annular and intermittent flow. The location of this boundary depends mainly on gas quality *x* and the mass flux *g_T_*. The vapor quality *x* at which intermittent flow changes into annular flow decreases as the mass flux of the fluid increases. At low mass flux, intermittent flow occurs when the vapor quality does not exceed 0.25, while at the highest mass flux (*g_T_* = 700 kg/(m^2^·s), annular flow already occurred when the vapor quality exceeded 0.05. Reference [19] demonstrated that annular flow in a channel filled with metal foam occurs for smaller values of vapor quality than in an empty channel.

Zhao, Lu and Tassou [20] investigated two–phase flow and boiling heat transfer in horizontal metal foam filled tubes (20 and 40 PPI). Plug, slug/wavy, stratified−wavy and annular flow patterns were indirectly judged through monitoring the cross−sectional wall surface temperature fluctuations and wall−refrigerant temperature difference. The experimental results show that the heat transfer was almost doubled by reducing cell size from 20 PPI to 40 PPI for a given porosity, due to extended surface area and intensive flow mixing in smaller cells.

Flow boiling heat transfer characteristics of R410A refrigerant in tubes filled with 5 and 10 PPI copper foam were experimentally investigated by Zhu et al. [21]. The flow patterns, such as slug/wavy, stratified−wavy, annular flow, slug and stratified flow, were observed. Both References [20] and [21] described the conditions corresponding to the occurrence of particular flow patterns in flow maps developed in the *x–g_T_* system. According to Zhu et al. [21] slug/wavy flow occurs at a relatively low mass flux of the mixture, that is, when the vapor quality *x* is smaller than 0.4. When *x* > 0.4, a transition to stratified or stratified−wavy flow occurs. Slug flow recognized for a high value of *g_T_* and small value of *x*, changes into annular flow for *x* > 0.4. An increase in the mass flow rate of fluid *g_T_* > 20 kg/(m^2^·s) results in the change of stratified flow into stratified−wavy flow. Annular flow was observed for *g_T_* > 90 kg/(m^2^·s). The experimental results reveal that metal foam enhances the flow boiling heat transfer by a maximum of 220% and promotes the flow pattern transition from stratified flow to stratified−wavy flow and from stratified−wavy flow to annular flow. Similar observations were made by Zhao, Lu and Tassou [20], who reported that the transition of intermittent flow (plug, slug and slug/wavy flow) into stratified or annular flow occurs when the vapor quality is smaller, that is, *x* exceeds 0.3. According to Zhu et al. [21], discrepancies of *x* values are attributed to the changes in physical properties of fluids applied in the experiments.

Some research, such as that in Reference [20], confirmed the influence of channel diameter and foam pore density on the area of occurrence of particular flow patterns. However, other studies such as that in Reference [22], concerned with the boiling of the same refrigerant during a flow through pipe with three times smaller diameter (7.9 mm), report that the increase in pore density of foam leads to a more likely formation of annular flow. This type of flow occurs with a smaller vapor quality and a lower mass flux compared to foams with a lower pore density. The authors of References [21] and [22] concluded that the presence of foam in the channel contributes to earlier formation of annular flow at hydrodynamic conditions typical for intermittent or stratified−wavy flow in empty channels.

Different conclusions are provided by Gao, Xu and Liang [23], who showed that a larger vapor quality is required to obtain annular flow in the channel containing a porous structure compared to the empty channel. We can note that this work was concerned with the flow boiling of R134a coolant in mini−channels containing foams with relatively high pore density of 110 PPI. Despite the small size of the channel and foam cells, the authors of Reference [23] recorded flow structures that are typical for channels with the dimension of a few millimeters. At the same time they indicate that in the case of bubbly flow, the vapor bubbles have larger diameters than in empty channels. This phenomenon can be explained by the fact that the foam structure prevents the bubbles from breaking off from the nucleation surface.

The latest reports into the structure of gas−liquid flow through metal foams include work of Li et al. [24]. The authors confirm the observations previously made by other researchers regarding the types of flow patterns formed in the channel packed with the metal foam with pore density of 20 PPI during boiling flow of R141b refrigerant. The flow pattern was visualized by high−speed imaging in the channel. At the same time, they note that the location of the transition lines corresponding to the occurrence of specific flow patterns on flow maps (plotted under different heat fluxes) depends not only on the mass flow and the vapor quality but also on the heat flux. The change in the heat flux affected the value of the vapor quality at which the transition of churn flow into slug/plug flow and then into annular flow takes place. The test results also showed that the flow patterns have a strong impact on the heat transfer coefficient and simultaneous dry−out phenomenon has a heat transfer deterioration effect. The comparison of heat transfer coefficient between empty channels and metal foam filled channels indicated that the heat transfer enhancement of metal foam is about 2.5 to 3 times greater at the mass flux of 100 kg/(m^2^·s) and 200 kg/(m^2^·s).

Insights into the current literature offer the conclusion that relatively little information on multiphase flows (especially with liquid component not be the refrigerant) through channels packed with metal foams is available with regard to such basic problems as flow patterns and flow regime maps. As a result of the poor understanding of the types of patterns and the conditions of their occurrence in porous structures, we are not able to adequately describe the significant effect of the flow pattern formation on the course and results of processes carried out by the application of multiphase mixtures. A variety of reports concerned with gas−liquid flow in metal foams more often indicate the benefits obtained by using foams instead of other types of packings or empty tubes [20,21], less often is information given with regard to the formation of specific flow patterns and its influence on the course and effectiveness of implemented process [24]. Therefore, it is difficult to discuss the issues concerned with the impact of fluid properties on flow patterns, their type and conditions in which they occur. Based on the current state-of-the-art, it is also impossible to clearly determine the impact of foam morphology (structure) on flow patterns.

Considering the above, the main objective of this work is to experimentally study the air−water and air−oil two–phase flows through packed channel with aluminum foams with porosity in the range of 0.92–0.94, description of hydrodynamic flow conditions at which the particular flow patterns occur, determination of influence of fluid properties on flow patterns as well as flow regime map development, which describes the area of occurrence of identified flow structures.

## 2. Experimental Study–Identification of Flow Patterns

The experimental tests involved the flow of two gas−liquid two–phase mixtures through channels with three types of porous structures comprising metal foam made from an aluminum alloy. In accordance with the manufacturer’s data, the foams used had a pore density of 20, 30 and 40 PPI (pore per inch). The aluminum foams with pore density of 20 PPI (AlSi7Mg) and 30 PPI (AlSi7Mg) were provided by m−pore GmbH while foam 40 PPI (Al 6101) was produced by ERG Aerospace Corporation and was available under the product trade name as DUOCEL^®^ aluminum foam (Figure 1a–c). Air, water and Velol−9Q machine oil were used as working fluids forming the two–phase mixture, the density, viscosity and surface tension of which were equal to *ρ_ol_* = 859.4 kg/m^3^, *μ_ol_* = 0.0027 Pa·s, *σ_ol_* = 0.046 N/m, respectively.

Pore density is the nominal size of foams declared by their producers. Foam structure is better characterized by other morphological parameters, including porosity *ε*, diameter of cells *d_c_* and pores *d_p_*. These parameters were determined based on the analysis of microscopic images of the foam skeleton. All foams were characterized by similar porosity and diverse cell and pore sizes. Forty PPI foam also had a different skeleton structure. The foam skeleton in many places creates clusters of solid material in the form of large nodes at the junction of skeleton fibers (Figure 1c). A summary of the most important morphological parameters of foams can be found in Table 1.

The observation of the flow patterns was carried out using an experimental installation, the main element of which was a horizontal channel section with an internal diameter of 0.02 m and a length of 2.61 m that was fully packed with foam (a separate channel was designed and built for each of the foams). Due to the fact that the installation was also used for heat exchange studies, the main section of the channel was made of aluminum pipe. Flow structures were observed in the final part of the channel that were 0.62 m long and made of transparent Plexiglas (Figure 2).

Air that was applied as the circulating medium was supplied to the experimental setup from a compressed air installation. Water was pumped using a multistage vortex pump and oil was pumped by a gear pump. Pump speed was controlled by varying its rotational speed and rerouting excess liquid (through a bypass system), which provided the required constancy of liquid flow rates. The flow rates of liquids were finally controlled by throttling valves and the values were measured by electronic flow meters. Mass flow meters were used to measure the air flow. The water flow was measured by means of a turbine−wheel flow meter and the oil flow by means of an oval gearwheel flow meter. The pressure and pressure drop in the installation was also measured due to the need to determine air density and identification of flow patterns based on measured pressure fluctuations. Piezoresistive pressure sensors and differential pressure sensors were applied for this purpose. The measuring equipment was calibrated before start of testing. Measurement uncertainty was assessed based on calibration results, in accordance with the guidelines of the International Organization for Standardization contained in the Guide to the Expression of Uncertainty in Measurement. The metrological data of the measuring apparatus are summarized in Table 2.

The tests were carried out at ambient temperature. Fluid temperature was 22–25 °C. Gauge pressure in the channel (section of flow patterns identification) did not exceed 62 kPa.

The identification of flow patterns was carried out based on visual observations and analysis of camera images and videos captured during the experiments as well as the measurement of the fluctuation of pressure drop (∆P) in time (t), between two measurement points located along the path of flow. The registration of the flow patterns was performed using a Canon 300 D digital camera and video camera. The camera used to take images of the flow patterns has a shutter speed of 1/4000 s, whereas the video camera shoots films at a resolution of 1024 × 1024 pixels and a frequency of 1800 Hz.

The research was carried out with a relatively wide range of variations of fluid mass flux *g_f_*. The volume ratios of air and liquid phases in the two–phase mixture were also varied to a considerable extent. The summary of the ranges of variations in the flow parameters is presented in Table 3. The superficial phase velocity *v_sf_* value was determined, taking into account the porosity *ε* of the foam using the equation,
(1)$$v_{sf}=Q_{f}\frac{4}{\varepsilon \pi d^{2}}$$
where *Q_f_*–volumetric flow rate of phase *f* (gas or liquid), m^3^/s; *d*–channel diameter, m.

The volume fraction of the phases *ζ_f_* was adopted to be represented by the volumetric flow rate of this phase in relation to the sum of gas and liquids rates the prior to their introduction into the channel,
(2)$$\xi_{f}=\frac{Q_{f}}{Q_{G}+Q_{L}}=\frac{Q_{f}}{Q_{T}}$$
where *f* = *L* (liquid phase) of *f* = *G* (gas phase). In the experiments using oil, the maximum fluid mass flux were smaller than during the air−water flow. Significant pressure drops accompanying air−oil mixture flow made it impossible to carry out the experiments with regard to this flow within the same range of parameters as in the air−water flow.

The observations of flow patterns were carried out in measurement series, during which a constant mass flux of liquid was carried through the channel, into which a specific air mass flux was supplied. When the flow was stabilized and the flow patterns were identified, the air mass flux was increased. In this way, structures were observed for 154 various air−water flow conditions and 95 for air−oil flow for each of the three foams.

## 3. Flow Patterns

Throughout the experiment, typical flow structures of the gas−liquid flow in the horizontal empty channel were observed. Stratified flow formed the most frequent pattern in the air−water flow. In addition, plug flow, semi−slug flow, slug flow, churn flow and annular flow were recorded. The latter type of flow did not occur in the air−oil flow, due to the smaller air mass flux compared to the case of the air−water flow tests. Diagrams of flow patterns recorded during the experiments are presented in Figure 3.

The types of flow patterns were determined primarily by the relation of the gas mass flux *g_G_* to the liquid mass flux *g_L_*. In the case of flows containing small liquid fluxes, the introduction of even a very small air mass flux into the channel resulted in the formation of stratified flow in which the liquid was carried along the bottom of the channel in the form of a continuous film (Figure 3a). The thickness of the liquid film depends on the fluid mass flux of mixture components introduced into the channel. The interfacial surface formed between the gas and liquid phases may be wavy to a varying degree, depending on the fluid velocity. The stratified flow occurred when the mass flux of gas *g_G_* 10 kg/(m^2^∙s) and the liquid mass flux did not exceed 70 kg/(m^2^∙s) for water and 30 kg/(m^2^∙s) for oil. For larger liquid mass flux, plug, slug and churn flow was observed. When a small amount of gas was introduced into a relatively large liquid mass flux, plug flow was formed. In this flow, liquid formed the continuous phase and the gas was carried in the form of elongated plugs in the top section of the channel (Figure 3b). Plug flow was observed when the superficial air velocity *v*_sa_ did not exceed 0.4 m/s and the ratio of liquid to gas velocities was at least 0.3. As the gas flow increased, the dimensions of the plugs increased. Their length was from about 2 to several centimeters. The largest plugs occupied around 3/4 of the channel cross−section.

Along with the increase in gas mass flux, plug flow changes into a slug flow. As the airflow increases, the plugs start to combine to each other. As a result, gas becomes the continuous phase. Due to the high air velocity, considerable wave structures are formed along the interface between the gas and liquid phases. Local accumulations of liquid are also noticeable in the channel. In extreme cases, accumulated liquid portions can even occupy the entire cross−section of the channel. Churn flow has a similar character and occurs in a similar range of fluid mass flux changes as slug flow but at larger air mass flux. Fast flowing gas causes a loss of liquid continuity, which flows in the form of aerated portions with an irregular shape. Portions of this mixture flow through the channel at high speed, cyclically at intervals of several seconds. Churn flow has not yet been observed by other researchers in foam packed channels.

Slug and churn flows are very unstable. In the case of these flows, pulsations of pressure associated with high fluid velocities and their dynamic changes in the distribution in the channel are observed. When gas flows at many times greater velocity than liquid in the channel, annular flow is formed (Figure 3e). In this flow, the liquid takes the form of a thin film flowing along the channel wall (and along its entire perimeter). Gas occupies the central part of the channel. Annular flow was recorded when the gas mass flux exceeded 7 kg/(m^2^·s), which corresponds to a superficial gas velocity of about 4 m/s

Along with the decrease of the liquid mass flux, this value increased to 10 kg/(m^2^·s). The results demonstrate that in the flow through the channels with foam packing, the gas velocity that is required to generate annular flow is several times smaller than in the case of flow through the horizontal channels without metal foam. In empty channels, annular flow usually occurs in the range of the gas velocities of about a dozen m/s. Due to its low viscosity, air can relatively easily flow through the foam, which decelerates water flow to a considerable extent and in turn this increases the interfacial slip that is typical for annular flow. Air−oil flow tests were carried out with a maximum air velocity of 2.5 m/s. This range of velocities did not provide sufficient conditions to form annular flow.

Under the same flow conditions, for air−oil and air−water flow, different types of flow patterns can be observed. For example, in Figure 4a, which deals with the air−water flow, the marked rectangle (black dashed line) represents the range of the experimental parameters changes for the air−oil flow. Inside this rectangle, only stratified flow and a few cases of plug flow occur in the air−water flow. However in the same flow conditions, besides these two flow patterns. in the air−oil flow the other flow regimes such as churn and slug flow can be observed.

This is even more clearly seen in Figure 5, in which both the results of air−water and air−oil tests are marked. In Figure 5, the graph coordinates *x* and *g_T_* describe mass fraction and mass flux of gas−liquid two–phase mixture, respectively. Mass fraction *x* is the ratio of gas mass flux (*g_G_*) to gas−liquid mass flux (*g_T_*). This figure clearly presents how plug and slug flow patterns overlap with stratified water flow. The oil covers the surface of foam better and has a higher viscosity than water, so a greater volume of oil content is held up in the channel. As a result, the air−oil intermittent flow (plug, semi−slug and slug) occurs at a smaller liquid mass fluxes than in the air−water flow.

The study did not demonstrate the effect of foam pore density and thus, cell size, on gas−liquid flow patterns. For all three foams, the same flow patterns were observed with the same fluid mass flux and their volume ratios in the mixture.

## 4. Flow Regime Map

The flow regime maps given in the literature with regard to metal foams differ quite significantly from one another. The areas corresponding to the occurrence of particular flow patterns presented on them do not conform to the results of the present experiment, as visually demonstrated in Figure 5. All types of flow observed in the experiments were recorded in conditions in which, according to the report in Reference [22], plug or slug flow patterns were formed. In their work, intermittent flows occur when the gas quality is less than 0.4, regardless of fluid mass flux. In turn, according to Li et al. [24] intermittent flow occurs when the gas content does not exceed 0.03–0.07. The exact transition lines between intermittent and annular flow in the paper [24] are attributed to the increased heat flux. In addition, as concluded in Section 3 of this paper, various air−water and air−oil flows may occur under similar conditions. The effect of the type of liquid on flow pattern formation implies that gas−liquid flow maps through metal foams should be developed taking into account the physical properties of fluids. Currently, such maps are known only for gas−liquid flows through empty channels. Examples include the Baker map [25] and the map by Mandhane et al. [26], developed in coordinate systems based on flow conditions and dimensionless parameters to account for the relations of physical properties of gas and liquid and the properties of air and water phases. Figure 6 and Figure 7 present the results of the described research against the background of these two maps. Parameters *X* and *Y* on the map of Mandhane et al. [26] are defined by the following relations,
(3)$$X={\left(\frac{\rho_{G}}{1.21}\right)}^{0.2}{\left(\frac{\rho_{L}}{1000}\right)}^{0.25}{\left(\frac{0.0728}{\sigma_{L}}\right)}^{0.25}{\left(\frac{\mu_{G}}{1.8\times 10^{-5}}\right)}^{0.2}$$
(4)$$\;Y={\left(\frac{\mu_{L}}{0.001}\right)}^{0.2}{\left(\frac{\rho_{L}}{1000}\right)}^{0.2}{\left(\frac{0.0728}{\sigma_{L}}\right)}^{0.25}$$
where *λ* and *ψ* are defined in the Baker’s map [25] in the following manner,
(5)$$\lambda ={\left[\left(\frac{\rho_{G}}{\rho_{a}}\right)\left(\frac{\rho_{L}}{\rho_{w}}\right)\right]}^{0.5}$$
(6)$$\psi =\frac{\sigma_{w}}{\sigma_{L}}{\left[\left(\frac{\mu_{L}}{\mu_{w}}\right){\left(\frac{\rho_{w}}{\rho_{L}}\right)}^{2}\right]}^{1/3}$$

In the case of both maps, the areas corresponding to the occurrence of individual flow patterns for flow through metal foam do not coincide with the corresponding regions found on the original maps (for flow in empty channels), which is consistent with the observations of other researchers, including the results reported in References [19] and [21].

Generally, we can state that in the case of channels with metal foam packings, the transformation of stratified flow into other types of flow occurs in the conditions when the ratio of the liquid flux to gas flux is smaller than in empty channels. This is particularly evident when the stratified flow changes into plug flow. This is due to greater interfacial slip accompanying flow through the channels packed with metal foams. As demonstrated in Reference [27], the volume fraction of liquid in the mixture pumped through foams is greater than for the case of flow through empty channels.

Despite the demonstrated incompatibility of the test results with flow maps through empty channels, a decision was made to develop a map for the flow through open–cell foams in the system proposed by Baker [25]. The application of this system has eliminated the overlap of areas accompanied by the formation of different air−water and air−oil flow patterns. The points applied to denote individual types of flow developed in the Baker system form compact regions and transition lines can be plotted between them. The updated map is presented in Figure 8. The identification of patterns during research was very difficult in many cases, due to the presence of foam in the channel; and for this reason, other flow patterns than the indicated ones can occur on the map in the immediate vicinity of the transition lines. When the map is applied to regions that lie beyond the range of experimental points, one should be aware of the possibility associated with the formation of other flow patterns than the ones that are described in this paper.

## 5. Conclusions

The type of flow pattern may determine the proper work conditions of the industrial apparatus, influence the efficiency of the processes carried out and determine the operating costs of whole installations. From a practical point of view, the knowledge of the hydrodynamic conditions under which particular flow patterns occur especially in conditions in which one flow pattern changes into the other one is of a great importance in many technological processes due to development of calculation procedures applied to pressure drops, void fraction, heat and mass transfer coefficients calculations.

There is a clear similarity in gas−liquid flow through channels with a porous structure made of metal foams to the flow through empty channels. In the reported studies, plug, slug, churn, stratified and annular flow were observed. Churn flow during flow in channels packed with open–cell foams has not yet been observed by other researchers. However, the flow through metal foams is accompanied by the formation of specific types of flow patterns that occur in different conditions than in empty channels. For this reason, gas−liquid flow maps through the empty channels cannot be used to predict flow patterns in channels filled with metal foams.

The type of flow pattern is determined by the conditions (velocity and volume fractions of phases) and fluid properties. The influence of the geometrical parameters of foams on flow patterns was not noted. In the flow through metal foams of air and water, stratified flow forms the dominant type of flow. In the case of air flow with oil, that is, a liquid with a significantly higher viscosity, a greater variety of flow patterns was observed.

Information derived from the literature, as well as their comparison with the results of experimental investigation conducted for open–cell aluminum foams with porosity in the range of 0.92−0.94, demonstrate the existence of some discrepancies regarding the location of the transition lines representing the boundaries of individual flow patterns occurring in various two–phase mixtures. This applies in particular to the transition line between stratified and annular flow. The reason for these discrepancies is the different physical properties of two–phase mixtures (and their components) used in the present research and other researchers’ work. The description of the areas of occurrence of flow patterns requires that the physical properties of fluids are taken into account. The research results in this area indicate the special role of liquid viscosity. The maps for gas−liquid two–phase flow through open–cell metal foam published so far have been developed in coordinates such as gas velocity−liquid velocity and vapor quality–mixture mass flux. Due to the use in the present experimental tests, two liquids that is, water and oil with different properties as two–phase mixture components, it was possible to observe a clear influence of the fluid properties on the type of flow patterns formed. Other researchers, as highlighted in the literature review, are focused mainly on the boiling of selected refrigerants, which did not allow such observations to be made. Thus, the maps for the boiling of refrigerants have limited application and can be used only for gas−liquid mixtures with specific properties.

It was demonstrated in this study that gas−liquid flow patterns accompanying flow through metal foams can be predicted by the application of the reference system proposed by Baker [25]. In this system, new transition lines were plotted with the purpose of identifying the areas where the individual types of flow occurred. A comprehensive flow regime map was developed based on approximately 250 experimental points covering a wide range of changes of flow parameters for gas−liquid two–phase systems in channels packed with open–cell metal foams where there is no change in the physical state of the mixture components (no evaporation process). The map covers the cases of two–phase flows and flow conditions that have so far been insufficiently studied and need further experimental research. A new developed map, compared to Baker’s map coordinates (*g_L_*/*g_G_*)*λ**ψ* versus *g_G_*/*λ* is valid for abscissa in the range of 0.4−6000 and ordinate in the range of 0.03−20, respectively. The range of these values on the Baker’s map correspond to the following changes in superficial velocities of gas (0.031−8.840 m/s), water (0.006–0.119 m/s) and oil (0.006–0.066 m/s) in two–phase mixture flow. A new flow map, unlike others available in the literature flow maps for boiling refrigerants, described two different gas−liquid systems, that is, air−water and air−oil flows and is valid for liquids with a density equal to or lower than the density of water and with a viscosity several times greater than that of water.

## Figures and Tables

**Figure 1 materials-13-03254-f001:**
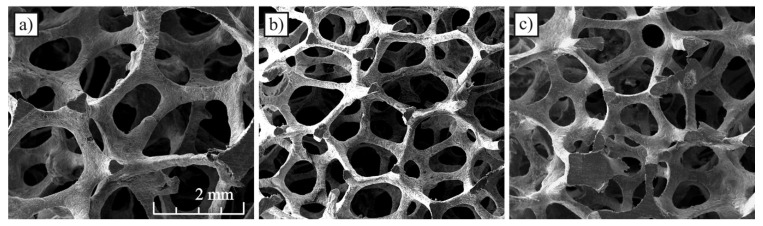
Open–cell aluminum foams applied in the experiments: (**a**) 20 PPI, (**b**) 30 PPI, (**c**) 40 PPI.

**Figure 2 materials-13-03254-f002:**
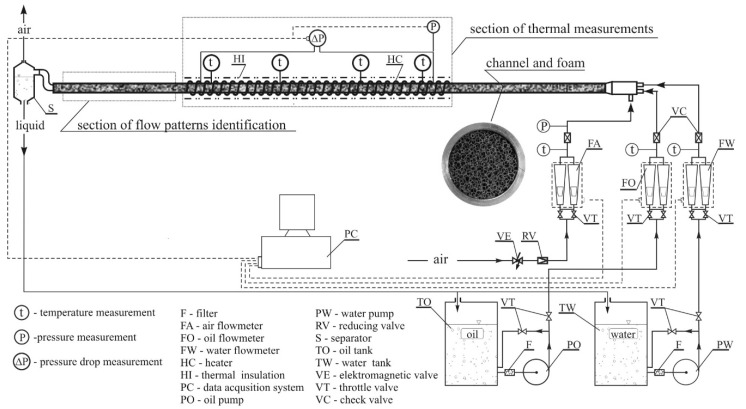
Experimental setup.

**Figure 3 materials-13-03254-f003:**
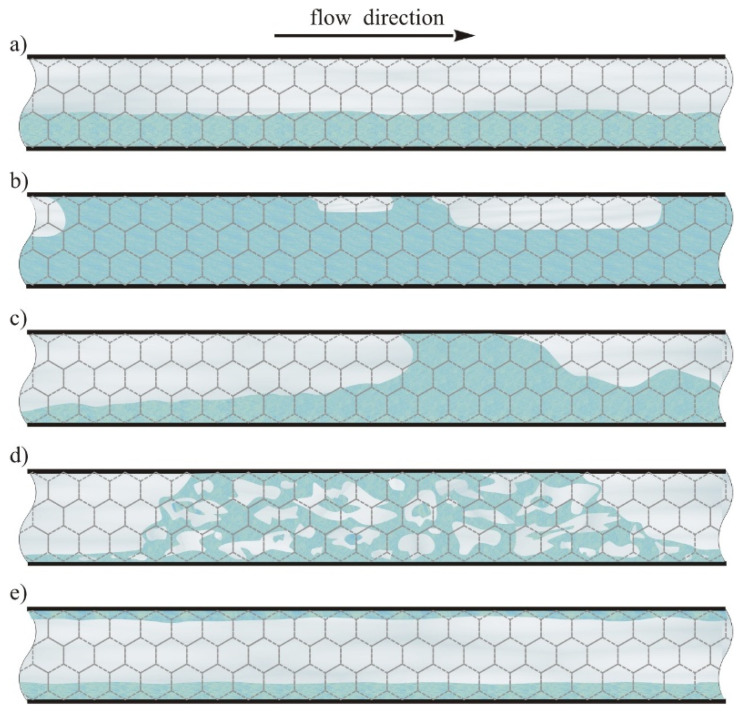
Gas−liquid flow patterns accompanying flow through metal foams: (**a**) stratified, (**b**) plug, (**c**) slug, (**d**) churn, (**e**) annular.

**Figure 4 materials-13-03254-f004:**
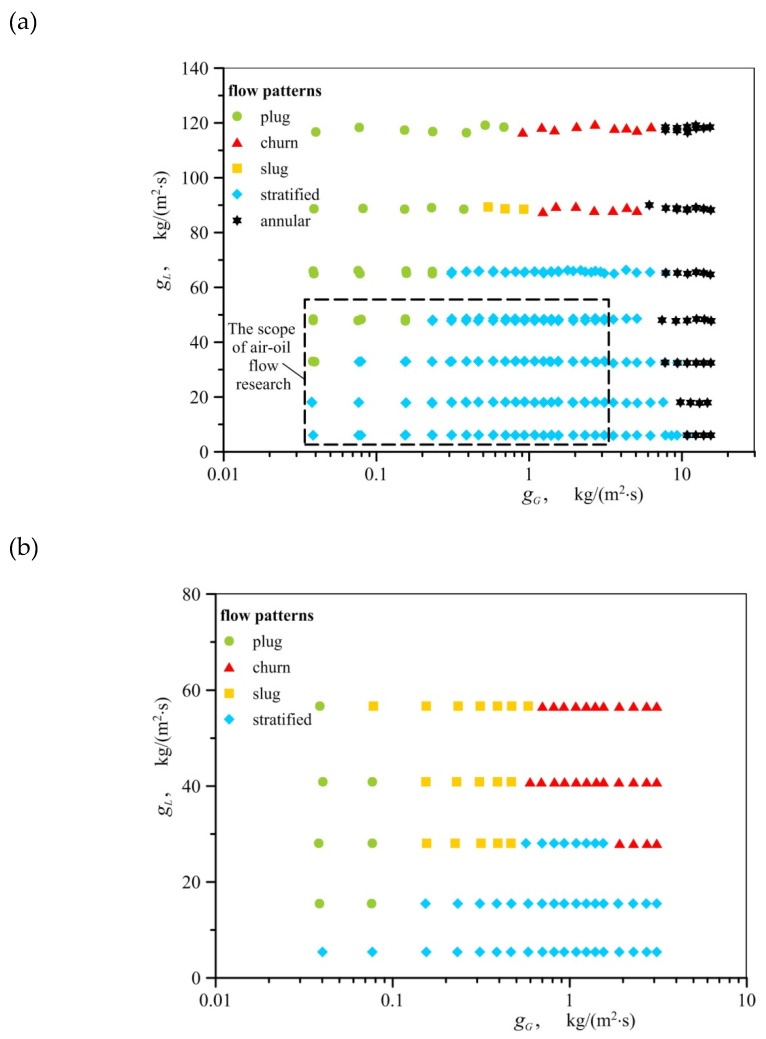
Conditions corresponding to occurrence of gas−liquid flow patterns: (**a**) air−water flow, (**b**) air−oil flow.

**Figure 5 materials-13-03254-f005:**
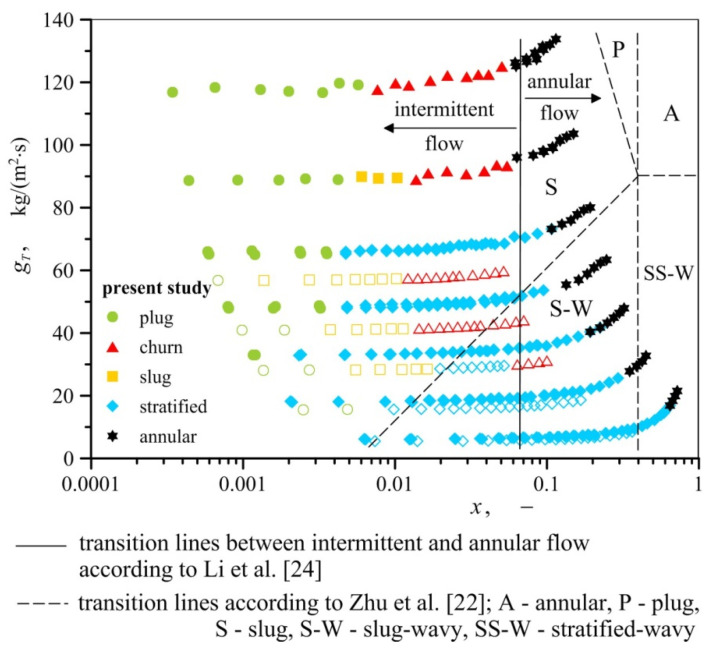
Air−water and air−oil flow patterns represented in the *x*–*g_T_* system (empty symbols refer to air−oil flow).

**Figure 6 materials-13-03254-f006:**
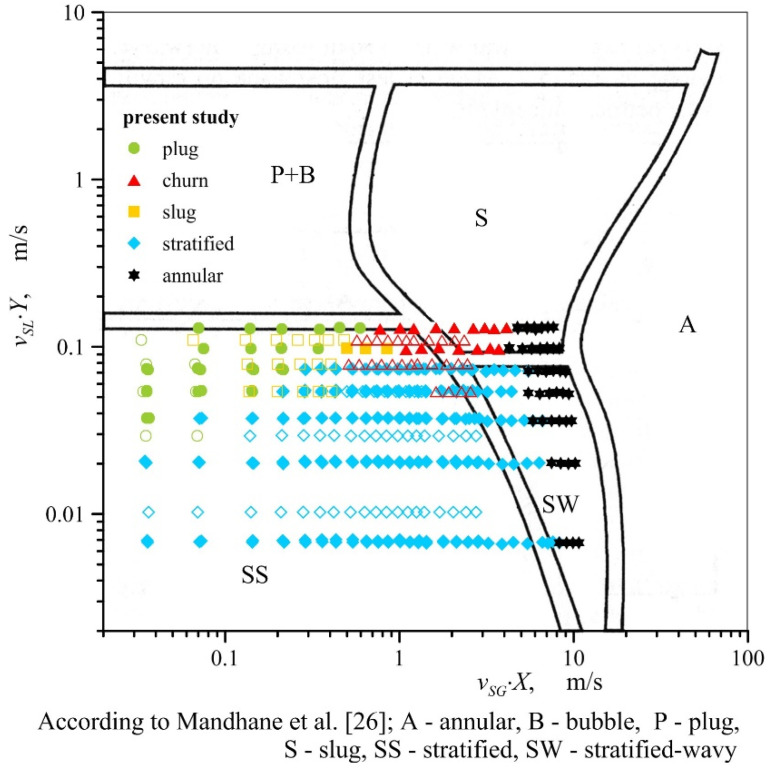
Ranges of pattern formation during gas−liquid flow through channels with metal foam packings vs. map developed by Mandhane et al. [26] (empty symbols refer to air−oil flow).

**Figure 7 materials-13-03254-f007:**
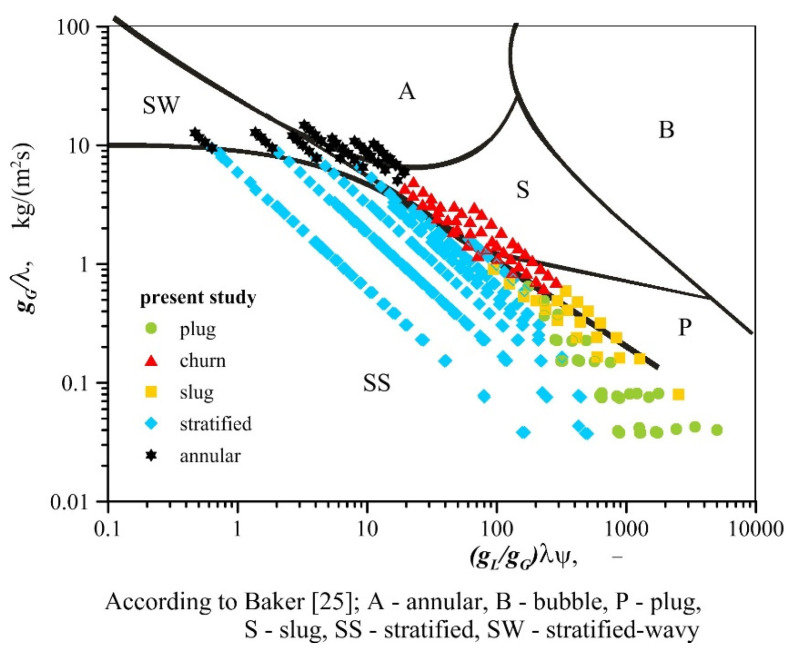
Ranges of pattern formation during gas−liquid flow through channels with metal foam packings vs. map developed by Baker [25].

**Figure 8 materials-13-03254-f008:**
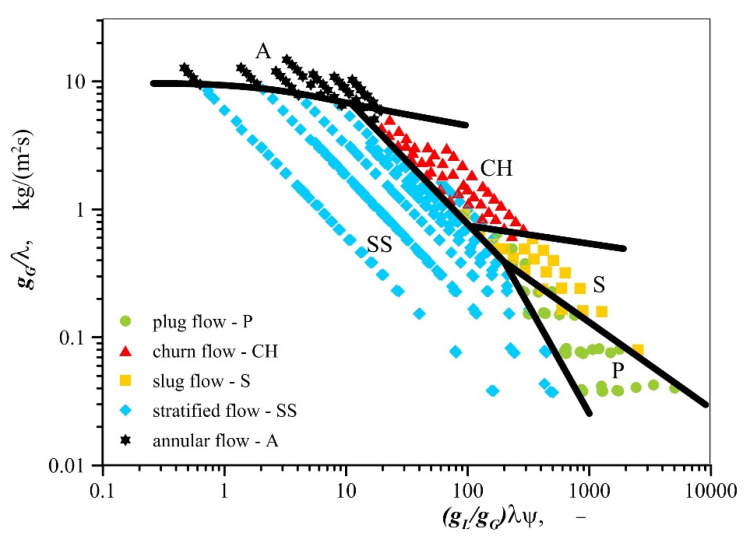
Gas−liquid flow map for horizontal flow through open–cell metal foams.

**Table 1 materials-13-03254-t001:** Characteristics of aluminum foams.

Parameter	Pore Density		
	20 PPI (AlSi7Mg)	30 PPI (AlSi7Mg)	40 PPI (Al 6101)
porosity *ε*, %	93.4	94.3	92.9
cell diameter *d_c_*, m	3.45 × 10^−3^	2.25 × 10^−3^	2.38 × 10^−3^
pore diameter *d_p_*, m	1.09 × 10^−3^	0.71 × 10^−3^	0.82 × 10^−3^

**Table 2 materials-13-03254-t002:** Specification of measuring equipment.

Measure	Sensor	Measurement Range	Accuracy *	Relative Uncertainty **
water flow rate	Kobold			
	DPM1107G2L	2.5 × 10^−7^–1.17 × 10^−5^ m^3^/s	1%	4.1%
	Kobold			
	DPM1550G2L	8.33 × 10^−6^–8.33 × 10^−5^ m^3^/s	1%	4.2%
air flow rate	Kobold			
	DMS111C4FD2	0–3.33 × 10^−4^ m^3^/s	0.2%	2.4%
	Kobold			
	DMS214C4FD2	3.33 × 10^−4^–3.33 × 10^−3^ m^3^/s	0.3%	1.7%
oil flow rate	Kobold			
	KZA 1804R08	3.33 × 10^−7^–6.67 × 10^−5^ m^3^/s	2%	6%
differential pressure	Aplisens PR−28	0–2000 Pa	1.6%	3.7%
	Aplisens PR−28	2–10 kPa	0.4%	1.2%
	Aplisens PR−28	10–50 kPa	0.25%	0.7%
	Aplisens PR−28	50–150 kPa	0.25%	0.4%
pressure	Aplisens PR−28	0–600 kPa	0.25%	0.3%
temperature	K−type thermocouples	0–100 °C	0.1 K	0.9%

* declared by the producer in relation to the measuring range. ** refers to the measured value determined on the basis of calibration.

**Table 3 materials-13-03254-t003:** Range of experimental data.

Fluid *f*	*v_sf_*, m/s	*g_f_*, kg/(m^2^·s)	*ζ_f_,* −
air, *a*	0.031–8.840	0.039–15.58	0.207–0.999
	0.031–2.550 *	0.039–3.11 *	0.319–0.998 *
water, *w*	0.006–0.119	5.99–119.49	0.001–0.793
oil, *ol*	0.006–0.066	5.40–56.71	0.002–0.681

* applied to air−oil flow.

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
