# Peer review of "Influence of Hydrodynamic Conditions on the Type and Area of Occurrence of Gas–Liquid Flow Patterns in the Flow through Open–Cell Foams"

_materials, 2020, doi:10.3390/ma13153254_

Round 1
Reviewer 1 Report
see pdf

Author Response
The responses to Reviewer comments are enclosed as attachment.

Reviewer 2 Report
The reviewer comments of the paper «Influence of hydrodynamic conditions on the type and area of occurrence of gas-liquid flow patterns in the flow through open-cell foams»
- Reviewer
The authors presented an article «Influence of hydrodynamic conditions on the type and area of occurrence of gas-liquid flow patterns in the flow through open-cell foams». In general, the article is written well and clearly. However, there are several points in the article that require further explanation.
Comment 1:
Demonstrate in the abstract novelty, practical significance. Give quantitative and qualitative of the proposed method in comparison with existing studies.
Comment 2:
The whole introduction is well written. However, citations and references must be drawn up in accordance with the requirements of the MDPI.
At the end of the introduction, give a brief summary of what has been done in each section. Give the purpose of research in this article.
Comment 3:
In Section 3, it would be good to give the structure as a percentage of porous structures, possibly in the table.
Comment 4:
Are the formulas in the article original? If not, then you need to quote.
Comment 5:
It will be useful to add a section of Nomenclature in which to sign all the physical quantities and abbreviations encountered in the article. There are many physical quantities in the text and such a section will help to find the description of the necessary element.
For example,
dp : Pore diameter (mm)
etc.
Comment 6:
In general, the conclusions are well written, but it is necessary to more clearly show the novelty of the article and the advantages of the proposed method. What is the difference from previous work in this area? Show practical relevance. It is necessary to give quantitative and qualitative indicators of the proposed method. What is the difference from other researchers. The conclusions should be consistent with the purpose of the article.
The article is interesting and relevant, but improvements are needed. After minor changes, the article may be considered for publication in journal "Materials".
Author Response

(The authors gave the same response as above.)

Reviewer 3 Report
This paper performs experimental investigations of gas-liquid flow patterns in open-cell foams. While the research topic is important, the manuscript is not recommended for publication unless the following remarks are addressed in a revised version.
1) Section 1 and Section 2 can be combined and re-written in a more concise manner. The current form looks tedious and the review of literature is not well-organized.
2) The validity of the present results is questionable, as the findings are mainly based on visual observations by human eyes, without any photos or video snapshots to support the study. The authors claim that “The human eye adapts better to insufficient lighting than optical image recording devices.” However, this sounds questionable, as state-of-art image and video capturing systems should be able to handle insufficient lighting conditions and provide good images. This is a key concern I have for this manuscript. If the authors cannot provide any photo image in support of their observations, they can consider adding some numerical simulations of the multiphase fluid flow within porous media, and if the simulation results agree with their experimental results, then the conclusions made in the manuscript will become more convincing.
3) The subscript f in Equation (2) is incorrect. It cannot be used as a dummy index of Q within a summation operator, if it is already used to denote a specific fluid type for Q.
4) How is Figure 3 plotted? Is it an illustration figure drawn by hand? Or is it a scientific figure based on physical experiment data?
5) Some English writing typos exist in the manuscript. For instance, “temperatur” should be “temperature” in Table 2.
Author Response

(The authors gave the same response as above.)

Round 2
Reviewer 2 Report
Comments are satisfied. The article can be published in journal "Materials".
Reviewer 3 Report
The authors have addressed my concerns about the manuscript. I suggest accepting it in the current form.